# Attitude of health professionals towards COVID-19 vaccination and associated factors among health professionals, Western Ethiopia: A cross-sectional survey

Tadesse Tolossa[1]*, Bizuneh Wakuma[2], Ebisa Turi[1], Diriba Mulisa[2], Diriba Ayala[3], Getahun Fetensa[2,4], Belayneh Mengist[5], Gebeyehu Abera[6], Emiru Merdassa Atomssa[1], Dejene Seyoum[1], Tesfaye Shibiru[7], Ayantu Getahun[1]

1 Department of Public Health, Institute of Health Science, Wollega University, Nekemte, Ethiopia, 2 Department of Nursing, Institute of Health Science, Wollega University, Nekemte, Ethiopia, 3 Department of Midwifery, Institute of Health Science, Wollega University, Nekemte, Ethiopia, 4 Department of Health Behavior and Society, Institute of Health, Jimma university, Jimma, Ethiopia, 5 Department of Public Health, College of Health Science, Debre Markos University, Debre Markos, Ethiopia, 6 Nekemte Health Center, Nekemte, Ethiopia, 7 School of Medicine, Institute of Health Sciences, Wollega University, Nekemte, Ethiopia

* yadanotolasa@gmail.com

## Abstract

### Introduction

Even though people of the world were eagerly waiting for the hope of vaccine development, vaccine hesitancy is becoming the top concern in both developed and developing countries. However, there is no adequate evidence regarding the attitude and perception of health professionals towards the COVID 19 vaccine in resource-limited settings like Ethiopia. The aim of this study was to assess health professionals' attitudes and perceptions towards COVID 19 vaccine in Western Ethiopia.

### Methods

An institution-based cross-sectional study was conducted among health care workers found in Nekemte town from April 14–21, 2021. A total of 439 health professionals present on duty during the study period was included in the study. The data were collected by using self-administered questionnaire. Epidata version 3.2 was used for data entry, and STATA version 14 was used for data analysis. The binary logistic regression model was employed to determine factors associated with the attitude towards COVID-19 vaccination. Adjusted Odds Ratio (AOR) with 95% confidence intervals was computed and statistical significance was declared at a 5% level (p-value < 0.05).

### Result

A total of 431 health professionals participated in the study yielding a response rate of 98.1%. The results indicated that 51.28% (95%CI: 45.12%, 57.34%) of health professionals had a favorable attitude towards COVID-19 vaccination. Having good knowledge about the COVID-19 vaccine (AOR = 0.38, 95%CI: 0.22, 0.64, P-value <0.001) was negatively

**Data Availability Statement:** All relevant data are within the paper and its Supporting information files.

**Funding:** The author(s) received no specific funding for this work.

**Competing interests:** The authors have declared that no competing interests exist.

**Abbreviations: AOR**, Adjusted Odds Ratio; **COR**, Crude Odds Ratio; **HCW**, Health Care Worker; **MLS**, Medical Laboratory Science; **WURH**, Wollega University Referral Hospital.

associated with unfavorable attitude towards COVID-19 vaccine, whereas age less than 30 years (AOR = 2.14, 95%CI:1.25,3.67, P-value <0.001), working in a private clinic (AOR = 7.77, 95% CI: 2.19, 27.58, P-value <0.001) and health center (AOR = 2.45, 95%CI: 1.01, 5.92, P-value = 0.045) were positively associated with unfavorable attitude towards COVID-19 vaccine.

## Conclusion and recommendation

In general, the attitude and perception of health care professionals toward the COVID-19 vaccine in the study area were unsatisfactory. Knowledge about the COVID-19 vaccine, age of health care workers, and place of work are the factors which affects attitude towards COVID-19 vaccine. Thus, we recommend the media outlets and concerned bodies to work to develop trust among the public by disseminating accurate and consistent information about the vaccine.

## Introduction

Starting from the day it was declared a pandemic, COVID-19 remains the worst Global public health challenge. According to the worldometer report, COVID-19 affects about 220 countries and territories. More than 350 million cases, and 5.6 million deaths happened due to COVID-19 as of January 24, 2022 [1,2]. The pandemic brought the double burden in developing countries already overwhelmed by the health care system challenges [3].

Given that sub-Saharan Africans are not the highest shareholder by cases and death during the early phase [4], the direct effect of COVID-19 and the indirect effect of its mitigation, disrupted the health care services [5]. The serious preventive measures (movement restriction, physical distancing, lockdowns, hand washing and sanitizing) were practiced [6]. However, the counter effect of some mitigation brought significant change in health care settings especially by task shifting and task sharing. Its effect on the economy is also evident. Though these mitigation procedures played a paramount role in averting the burden of COVID19, the whole world was hoping for vaccine development [7].

A few COVID-19 vaccines that are being used globally or locally [7]. As of June 10, 2021, from 287 candidate vaccines, 102 are in the clinical phase, 185 are in the pre-clinical phase [8]. From these, WHO has listed the Pfizer/BioNTech, Astrazeneca-SK Bio, Serum Institute of India, Janssen and Moderna vaccines for emergency use [6,7]. Believing Health Care Workers (HCWs) are explicitly exposed to increased risk of infection through direct contact with patients, they should be prioritized for vaccination [9].

The willingness of the general population to accept the vaccine ahead of COVID-19 vaccine development was relatively promising compared to today's reality [10–12]. Even though people of the world were eagerly waiting for the vaccine development, vaccine hesitancy is becoming the top concern in both developed and developing countries [11,13–17]. For instance, the vaccine acceptance is 21% in Egypt [18], 54.6% in China[19], and 57.6% of the adult population in the USA [20] intended to be vaccinated. Studies conducted in different regions of Ethiopia reported willingness to take COVID-19 vaccine were 46.1% in Southern Ethiopia [21],39.7 in Addis Ababa [22], and 48.6% in Southwestern Ethiopia [23]. The most common reason mentioned for hesitancy were concerns about the safety of a vaccine and wide-ranging lack of confidence, worries about the efficiency of the vaccine[17,18,24]

Several studies indicate that healthcare professionals (HCPs) play a paramount role and can significantly affect the general public's decisions to receive the COVID- 19 vaccine [25,26]. In Ethiopia, there have been 169,640 confirmed cases of COVID-19, with 9651 deaths till May 30, 2021 [1]. The Ethiopian government and the Ethiopian Ministry of health exert great efforts to provide the COVID-19 vaccines and sort the vaccination as a priority for healthcare workers (HCWs) and older people, especially people with chronic diseases history [27]. To overcome the expected upcoming challenge of vaccination hesitancy, we have to measure and know the exact reasons. However, there is no adequate evidence regarding the attitude and perception of health professionals towards the COVID-19 vaccine in resource-limited settings like Ethiopia. Hence, this study aimed to assess health professionals' attitudes and perceptions towards COVID 19 vaccine in West Ethiopia.

## Methods

### Study area and period

This study was conducted in health institutions located in Nekemte town. Nekemte is the capital city of East Wollega Zone, and it is located 330 KM from Addis Ababa, the capital city of Ethiopia. The study was conducted from April 15–21, 2021. Data from Nekemte Town Health Office showed that the town has one teaching referral hospital owned by Wollega University, one specialized hospital administered under Oromia Regional Health Bureau, two health centers (Nekemte and Cheleleki Health center), and more than fifteen medium and above private clinics, and more than 800 health professionals are found in Nekemte town.

### Study design

An Institutional based cross-sectional study design was employed.

### Population, sample size and sampling techniques

All health professionals working in private and public health institutions of Nekemte town were a source population. Health professionals who were not on duty due to different reasons were excluded from the study. All health professionals on duty during data collection and willing to participate in the study were selected.

The sample size was determined by single population proportion with the following assumptions: Since this study was the first of its type in Ethiopia, p = 50% was taken, with a 5% margin of error and a 95% confidence level. Accordingly, the calculated sample size was 384, and after adding a 15% allowance for a non-response rate, the final sample size was 442 health professionals.

All health institutions found in Nekemte town were included in the study. There are two public hospitals in Nekemte town, two health centers, and 15 medium clinics. Then the sample size was proportionally allocated to hospitals, health centers and private clinics. Around 710 health professionals are working in two hospitals, 50 in two health centers and 60 health professionals in all private clinics. For hospitals, we have used the identification number of the health professionals, and the identification number was used as a sampling frame. Then computer-generated simple random sampling technique was used to select the sample. Health professionals who were not on duty during the study period were excluded from the sampling frame. For health centers and private clinics, all health professionals who were available during the study period were included in the study. Furthermore, health professionals working in more than one health facility were considered only in a single health facility to prevent any distortion of information.

## Variables

The attitude of health professionals towards the COVID-19 vaccine was the dependent variable of this study. For attitude questions, the likert-scale method with a five points scale (strongly agree, agree, neutral, disagree, strongly disagree) responses were used to allow the study participants to express how much they agree or disagree with a particular question. Ten items were used to assess the attitude of health professionals towards COVID-19 vaccine. Participants' response was from 10 to 50. Higher scores denoted a "favorable attitude" towards COVID-19 vaccine. "Favorable attitude" was when the scoring was $\geq$ mean or 25 (50% and above) out of 50 items and $< 25$ ($<$50%) was rated as "unfavorable attitude".

Socio-demographic variables such as (age, sex, marital status, educational level, educational background, religion), medical history (chronic medical disease and previously infected with COVID-19), knowledge towards COVID-19 vaccine, and perception towards COVID-19 vaccine) were independent variables of this study.

For knowledge related items, the questions contained the category of ("Yes"/"No"). A correct answer was assigned "1" point and an incorrect answer was assigned "0" points. The total score ranged from 0 to 5. "Good knowledge" was when the scoring was $\geq$2.5 (50% and above) out of 5 items and score below 2.5 indicated "poor knowledge" on COVID-19 vaccine.

Finally, the perception of participants towards COVID-19 vaccine was assessed by using five items with "Yes"/No" category. The total score ranged from 0 to 5. Respondents who scored greater than or equal to the mean score ($\geq$ 2.5 or $\geq$ 50%) were grouped to have "good perception" and participants who scored less than the mean score ($<$2.5 or $<$50%) were grouped to have "poor perception" towards COVID-19 vaccine (S3 File).

## Data collection techniques and data quality assurance

A questionnaire was developed by reviewing previously published papers [28,29], and adapted to local context. The tool was designed and distributed to respondents in English language since the participants could read, write, and understand the language. The questionnaires comprised socio-demographic data, medical history, knowledge, attitude, and perception towards the COVID-19 vaccine. The data was collected by using self-administered questionnaires. The questionnaire was given to all health professionals on duty and returned to data collectors after filling it. Cronbach's alpha was used to assess the reliability of the tool, and the value was 0.78 (value more than 0.7 to 0.95 is acceptable) [30].

To ensure its quality, the questionnaire was pre-tested on 5% of participants. Then possible amendments were done based on the findings. The discussion was held between investigators and data collectors, based on the pre-test result, and accordingly, some amendments were made. The data collectors gave the one-day training about the tool and data collection procedure. Data were checked daily for completeness, accuracy, clarity, and consistency by the supervisors and principal investigator. Any error or ambiguity, and incompleteness were corrected accordingly.

## Data management and analysis

Epidata version 3.0 was used for data entry [31], and exported to STATA version 14.0 for further analysis [32]. Descriptive statistics, like frequencies, percentages, mean and standard deviation were computed. Before analysis, data were cleaned and edited by using simple frequencies and cross-tabulation. Re-categorization of categorical variables and categorization of continuous variables was done. The assumption of the logistic regression model was checked before fitting to the model. The binary logistic regression model was fitted to determine factors associated with attitude towards COVID-19 vaccine. The multivariable logistic

regression analysis included factors associated with the outcome variable at 20% (p-value ≤0.20) significant level in the bivariable logistic regression analysis. Then crude and adjusted odds ratio and their corresponding 95% confidence intervals were presented in the final multi-variable logistic regression table. Finally, AOR with 95% confidence intervals was computed and statistical significance was declared when it was significant at a 5% level (p-value < 0.05). Correlation matrices checked multicollinearity (association between explanatory variables), and the model goodness of fit test was checked by Hosmer and Lemeshow test.

### Ethical considerations

The study was approved, and ethical clearance letters were obtained from Wollega University, Institute of Health Science Research Ethics review board (**Min. No. 07/2021**). After approval, a permission letter was obtained from the administrative body of health facilities to respective clinics. Verbal consent was obtained from study participants, and the purpose of this study was stated to all participants. Everybody participated voluntarily in this study.

## Results

### Socio demographic characteristics of the health professionals

Four hundred thirty-one health professionals were filled the questionnaire and yielded a response rate of 98.1%. Two hundred sixty eight (62.28%) of participants were male. Regarding the age of the health professionals, nearly two-thirds of them, 274 (63.57%), belong to less than 30 years old age group. More than half 253(58.07%) of the participants were protestant religion followers. Greater than three fourth of the health professionals who participated in this study were from hospital institutions 350 (81.21%) (Table 1).

### Medical disorder related characteristics of the health professionals

Thirty-three healthcare workers have a history of taking vaccination in their lives (7.66%). Nearly one fifth of the health care workers in this study had chronic medical diseases 19 (4.41%). Participants' most commonly reported chronic disease was hypertension 6 (46.15%). Greater than half of the participants were vaccinated against coronavirus 240 (55.68%).

### The attitude of the health professionals towards COVID-19 vaccine

Almost half 210 (48.72%) of the participants have poor attitude toward COVID-19 vaccination. One-third of the participants strongly agreed that COVID-19 could not be controlled without vaccination. Moreover, nearly one-third of participants agreed that mass vaccination against COVID-19 helps overcome the pandemic (Table 2). Regarding the effectiveness of the newly discovered COVID-19 vaccination, less than a quarter of the participants strongly disagreed with 63 (14.62%) (Fig 1).

### Health care workers perception towards COVID-19 vaccine

The distributions of each perception item about the COVID-19 vaccine are presented in Table 3. Concerning the question "Do you think the COVID-19 vaccine is effective"? Nearly one-third of them thought the vaccine against COVID-19 was effective. More than half of the participants responded unsure for the question "Do you think COVID-19 will be controlled only by preventive measures without vaccination"? (Table 3).

Almost half of the participants accept that the newly developed COVID-19 vaccine has side effect 209 (48.49%) (Fig 2).

**Table 1. Socio demographic characteristics of the health professionals in Nekemte city, Western Ethiopia, 2021.**

| Variables | Option | Frequency | Percent |
|---|---|---|---|
| Age | <30years old | 274 | 63.57 |
| | > = 30years old | 157 | 36.43 |
| Sex | Male | 268 | 62.18 |
| | Female | 163 | 37.82 |
| Marital status | Never married | 176 | 40.84 |
| | Married | 243 | 56.38 |
| | Separated | 8 | 1.86 |
| | Others | 4 | 0.93 |
| Religion | Muslim | 73 | 16.94 |
| | Protestant | 253 | 58.70 |
| | Catholic | 10 | 2.32 |
| | Orthodox | 71 | 16.47 |
| | Others | 24 | 5.57 |
| Institution | Hospitals | 350 | 81.21 |
| | Health center | 34 | 7.89 |
| | Private | 47 | 10.90 |
| Education level | Diploma | 38 | 8.82 |
| | First degree | 362 | 83.99 |
| | Masters | 24 | 5.57 |
| | Doctors | 2 | 0.46 |
| | Specialty degree | 5 | 1.16 |
| Back ground | Nurses | 186 | 43.16 |
| | Midwifery | 56 | 12.99 |
| | Pharmacy | 32 | 7.42 |
| | Medical practitioner | 47 | 10.90 |
| | Public health | 41 | 9.51 |
| | Anesthetists | 19 | 4.41 |
| | MLS | 36 | 8.35 |
| | Psychiatrics | 6 | 1.39 |
| | others | 8 | 1.86 |

## Factors associated with attitude of health professionals towards COVID-19 vaccine

In multivariable logistic regression, variables like age, sex, institution types, having chronic diseases and allergic reaction to previous medication have shown significant association with the outcome of interest. From those variables, two of them showed significant association with the attitude of health professionals toward a vaccine against COVID-19. The odd of developing poor attitude toward COVID-19 vaccine was 2.14 times higher among health professionals aged < 30 years than their counterparts (AOR = 2.14, 95%CI: 1.25, 3.67). The likelihood of having a poor attitude toward the COVID-19 vaccine was 2.45 times higher among health professionals working at health centers than those working at hospitals AOR = 2.45(95%CI 1.01,5.92). Similarly, the odds of having a poor attitude towards the COVID-19 vaccine was 7.77 times higher among health professionals working at private clinics and hospitals AOR = 7.77(95% CI 2.19,27.58). Moreover, the proportion of health professionals with unfavorable attitudes was 62% lower among professionals who have good knowledge of COVID-19 vaccination than their counterparts AOR = 0.38(95%CI: 0.22, 0.64) (Table 4).

**Table 2. Attitude of the health care workers towards COVID-19 vaccine in Nekemte health facilities, Western Ethiopia.**

| Variables | Option | Frequency | Percent |
|---|---|---|---|
| The vaccine that is currently given in Ethiopia is the actual one that those innovative countries are taking | Strongly disagree | 114 | 26.45 |
| | Disagree | 132 | 30.63 |
| | Neutral | 81 | 18.79 |
| | Agree | 79 | 18.33 |
| | Strongly agree | 25 | 5.80 |
| If one person takes COVID-19 vaccination, it has a great contribution for other people | Strongly disagree | 65 | 15.08 |
| | Disagree | 86 | 19.95 |
| | Neutral | 53 | 12.30 |
| | Agree | 155 | 35.96 |
| | Strongly agree | 72 | 16.71 |
| I will take the vaccine if I get it without hesitation | Strongly disagree | 61 | 14.15 |
| | Disagree | 183 | 42.46 |
| | Neutral | 51 | 11.83 |
| | Agree | 86 | 19.95 |
| | Strongly agree | 50 | 11.60 |
| I encourage my family and others to take the vaccination | Strongly disagree | 58 | 13.46 |
| | Disagree | 135 | 31.32 |
| | Neutral | 61 | 14.15 |
| | Agree | 127 | 29.47 |
| | Strongly agree | 50 | 11.60 |
| COVID-19 cannot be controlled without vaccination | Strongly disagree | 85 | 19.72 |
| | Disagree | 136 | 31.55 |
| | Neutral | 65 | 15.08 |
| | Agree | 111 | 25.75 |
| | Strongly agree | 34 | 34 |
| COVID-19 vaccine is fairly distributed for all | Strongly disagree | 67 | 15.55 |
| | Disagree | 122 | 28.31 |
| | Neutral | 84 | 19.49 |
| | Agree | 99 | 22.97 |
| | Strongly agree | 59 | 13.69 |
| Mass vaccination can overcome the epidemic attack of the COVID-19 | Strongly disagree | 53 | 12.30 |
| | Disagree | 90 | 20.88 |
| | Neutral | 77 | 17.87 |
| | Agree | 138 | 32.02 |
| | Strongly agree | 73 | 16.94 |
| The best prevention method is to take vaccine against COVID-19 | Strongly disagree | 54 | 12.53 |
| | Disagree | 91 | 21.11 |
| | Neutral | 92 | 21.35 |
| | Agree | 134 | 31.09 |
| | Strongly agree | 60 | 13.92 |
| The COVID-19 vaccine is not tested adequately for its effectiveness | Strongly disagree | 28 | 6.50 |
| | Disagree | 73 | 16.94 |
| | Neutral | 67 | 15.55 |
| | Agree | 129 | 29.93 |
| | Strongly agree | 134 | 31.09 |
| After vaccinated against COVID-19 other precaution can be avoided | Strongly disagree | 89 | 20.65 |
| | Disagree | 131 | 30.39 |
| | Neutral | 62 | 14.39 |
| | Agree | 66 | 15.31 |
| | Strongly agree | 83 | 19.26 |
| The overall attitude towards COVID-19 vaccine | Favorable | 221 | 51.28 |
| | Unfavorable | 210 | 48.72 |

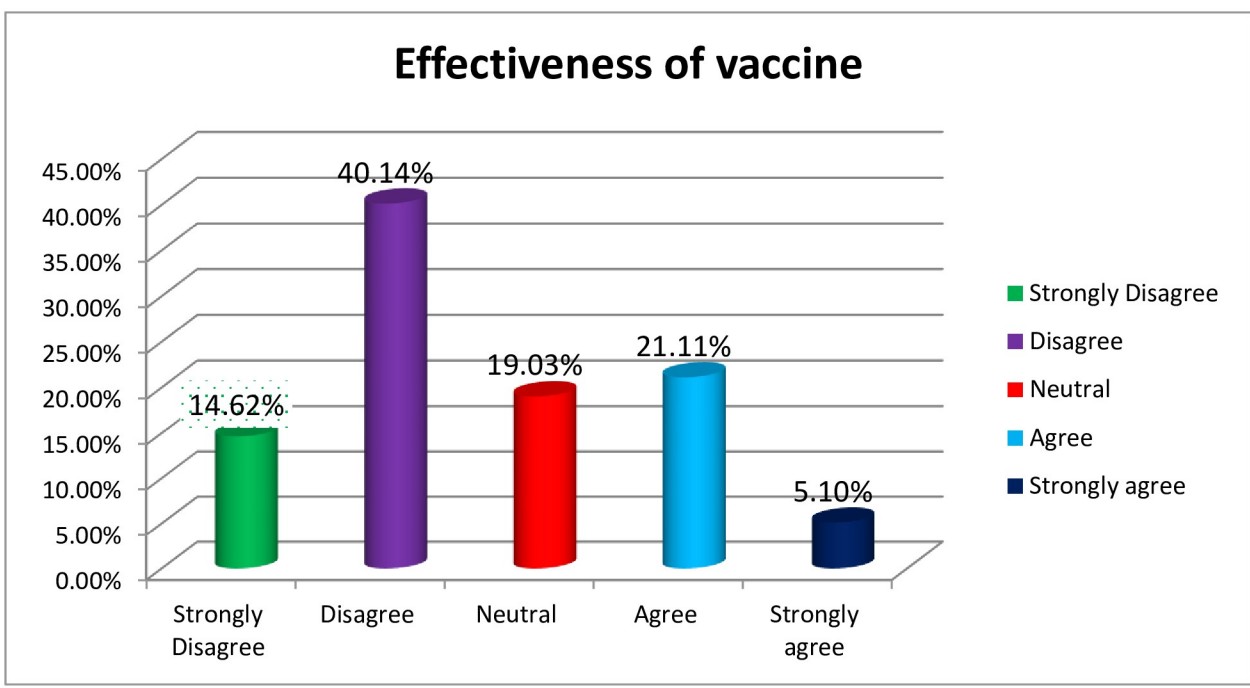

**Fig 1. Attitude of the health professionals regarding effectiveness of the Corona virus vaccine, 2021.**

## Discussion

Though the vaccine development against COVID-19 was promising for the world population, vaccine hesitancy has become a global challenge to the successful uptake of the vaccine [11,13–17]. Literature shows that vaccine safety and efficacy are among the worries of the people and the reasons for vaccine hesitancy [17,18,24]. Health care professionals are at a particular concern to be vaccinated to influence the general population for vaccination positively. There was a dearth of information about the attitude and perception of health professionals toward COVID-19 vaccination and its associated factor in Ethiopia, particularly in the study area. Therefore, this study was intended to determine the attitude of health professionals toward COVID-19 vaccination and its associated factors in Nekemte town.

Accordingly, 31.55% of the study participants had agreed to take the vaccine without hesitation if available in the present study. This is lower than the study findings from Southwestern Ethiopia (48.4%) [23], Eastern Ethiopia (61.4%) [33], Bangladesh (58.6%) [34], Canada (72.40) [35]. The possible explanation for this might be the variation in trust and reliability of the source of information about COVID-19 vaccine. Moreover, 41.07% of the study subjects in the present study have also agreed on encouraging families and others to take the vaccine while 65.5% of participants in Bangladesh agreed to do so. Furthermore, 59.75% of this study's health professionals agreed that COVID-19 would not be controlled without vaccination. This echoes the finding from Bangladesh (63.4%) [34]. In addition, nearly half of the study participants in the current study have agreed on mass vaccination to overcome the pandemic. This is also congruent with study done in Libya [36]. Comparable to the study finding from Saudi Arabia (37%) [37], only one-third of participants agreed on the vaccine's effectiveness in the present study. This depicts that there is still a need to provide reliable and accurate information about the vaccine's effectiveness against COVID-19 to health care professionals and the public.

**Table 3. Perception the health care professionals towards COVID-19 vaccine.**

| Variables | Option | Frequency | Percent |
|---|---|---|---|
| Have you ever been infected with COVID-19 | Yes | 95 | 22.04 |
| | No | 169 | 39.21 |
| | I don't know | 167 | 38.75 |
| Do you think that the COVID-19 vaccine is effective? | Yes | 131 | 30.39 |
| | No | 34 | 7.89 |
| | I don't know | 266 | 61.72 |
| Do you think that COVID-19 vaccine is mandatory for health care workers? | Yes | 262 | 60.79 |
| | No | 24 | 5.56 |
| | I don't know | 145 | 33.64 |
| Do you think COVID-19 will be controlled only by preventive measures without vaccination? | Yes | 85 | 19.72 |
| | No | 102 | 23.67 |
| | I don't know | 244 | 56.61 |
| Do you think that various COVID-19 vaccines have been discovered? | Yes | 142 | 32.95 |
| | No | 46 | 10.67 |
| | I don't know | 243 | 56.38 |
| Do you think that the COVID-19 vaccine will be affordable and accessible by the common person! | Yes | 59 | 13.69 |
| | No | 145 | 33.64 |
| | I don't know | 227 | 52.67 |

The current study revealed that half of the health care professionals who participated in this study have a favorable attitude toward COVID-19 vaccination. This is lower than the study done in Bangladesh reported that 78% of the general population had a favorable attitude [34]. However, it is higher than the online survey which was done in Ethiopia that reported 24.2%

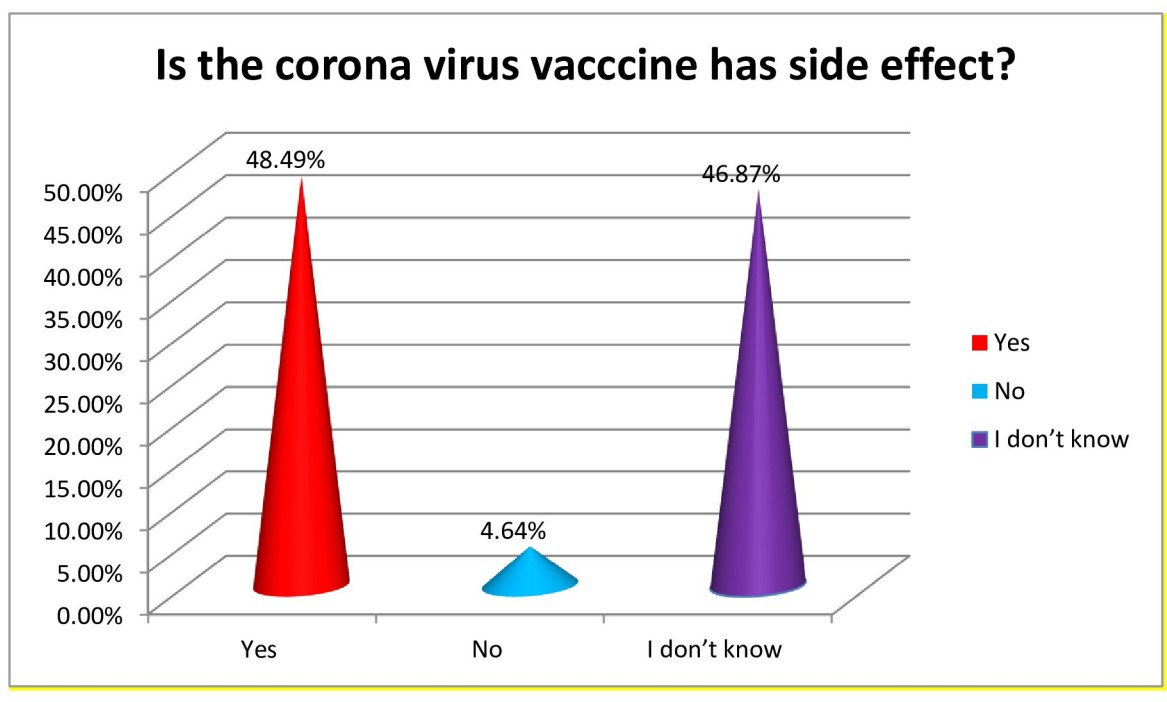

**Fig 2. Perception the health care workers regarding Corona virus vaccine's side effect, 2021.**

**Table 4. Multivariable analysis of factors associated with attitude towards COVID-19 vaccine among health professionals in Nekemte city, Western Ethiopia.**

| Characteristics | | Attitude | | COR(95%CI) | AOR(95%CI) | P-value |
|---|---|---|---|---|---|---|
| | | Poor | Good | | | |
| Age | <30 | 149 | 125 | 1.87(1.25,2.79) | 2.14(1.25,3.67) | <0.01* |
| | > = 30 | 61 | 96 | Ref | | |
| Sex | Male | 125 | 143 | Ref | | |
| | Female | 85 | 78 | 0.80 (0.54,1.18) | 0.93(0.54, 1.59) | 0.795 |
| Institution | Hospital | 195 | 155 | Ref | | |
| | Health center | 10 | 24 | 3.01(1.40, 6.50) | 2.45(1.01,5.92) | 0.045* |
| | Private clinic | 5 | 42 | 10.56 (4.0,27.35) | 7.77(2.19,27.58) | <0.01* |
| Chronic diseases | Yes | 9 | 10 | Ref | | |
| | No | 201 | 211 | 0.94(0.37,2.37) | 0.76(0.25,2.29) | 0.630 |
| History of allergic reaction to previous medication | Yes | 21 | 20 | Ref | | |
| | No | 189 | 201 | 1.11(0.58,2.12) | 1.04(0.46,2.33) | 0.914 |
| Knowledge about COVID-19 vaccination | Poor knowledge | 44 | 86 | Ref | | |
| | Good knowledge | 139 | 163 | 0.30(0.18,0.49) | 0.38(0.22,0.64) | <0.01* |

of the general population had a favorable attitude toward COVID-19 vaccination [38]. The possible reason for the observed discrepancy might be due to the relative information difference among the population of interest. Furthermore, it might be due to health care professionals being at higher risk of infection with the virus that might result in a favorable attitude toward vaccination to minimize the risk of infection. This implies a need to assure the vaccine's effectiveness, safety, and efficacy to enhance its uptake by health professionals and the general public. Health professionals are the counselor and advocators of the general population that their decision to receive the vaccine can greatly influence the uptake by the general population.

The factors affecting health care professionals' attitude toward COVID-19 vaccination were knowledge towards COVID-19 vaccination, age of respondents, and type of health facility. Accordingly, the proportion of health professionals who have unfavorable attitudes was 62% lower among professionals with good knowledge of COVID-19 vaccination than their counterparts. This might be because knowing the importance of the COVID-19 vaccine can positively influence someone to have a positive attitude toward it. Moreover, the odds of having unfavorable attitudes were much higher among health professionals working in private clinics and health centers than those in hospitals. This variation might be due to the relative number of client flow to the health center and private clinic being much lower than the hospital and, hence the professionals consider themselves at lower risk of exposure and infection to COVID-19. In addition, almost all COVID-19 patients have been admitted to hospitals than private clinics or health centers. Therefore, professionals working at health centers and private clinics for known and obvious reasons consider themselves to have a lower risk of infection with COVID-19. As a result, they might have an unfavorable attitude toward the COVID-19 vaccine than professionals working in the hospital setting.

In this study, age of HCWs were significantly associated with attitude towards COVID-19 vaccine, in which health professionals aged less than 30 years had negative attitude towards the vaccine. This is in line with study conducted in North Ethiopia which reported higher age positively associated with positive attitude towards COVID-19 vaccine [39]. This might be due to the fact that, as age increase, the probability of developing comorbidity increase, and the chance of infecting with COVID-19 also high. Thus the HCWs intention to receive the vaccine would be high.

### Limitation of the study

The study did not provide qualitative perspectives on the attitude and perception of health professionals towards the COVID-19 vaccine and its associated factors, which, if available, could underpin the quantitative findings. Moreover, since it is a cross-sectional study, it did not address the cause and effect relationship between the factors and the outcome variables.

## Conclusion

In general, the attitude and perception of health care professionals toward the COVID-19 vaccine in the study area were unsatisfactory. Having poor knowledge about the COVID-19 vaccine, young age group, working in private clinics and health centers are the independent determinants of unfavorable attitudes towards the COVID-19 vaccine. Therefore, there is still a need to improve health professionals' knowledge of the COVID-19 vaccine by providing reliable information regarding vaccine safety, efficacy, and effectiveness. Furthermore, the media outlets need to work to develop trust among the public by disseminating accurate and consistent information about the vaccine. In addition, future researchers should also explore more about the attitude and perception of health care professionals toward the COVID-19 vaccine and its determinants using qualitative data.

## Supporting information

**S1 File. Dataset.**
(DTA)

**S2 File. Strobe checklist.**
(DOCX)

**S3 File. Tool.**
(DOCX)

## Acknowledgments

We want to thank all health facilities for their invaluable co-operation during data collection, and our deep acknowledgment also goes to the data collectors for their interest and commitment in carrying out the study.

## Declaration

**Ethical approval and consent to participate**. The study was approved, and ethical clearance letters were obtained from Wollega University, Institute of Health Science Research Ethics review board. After approval, a permission letter was obtained from the administrative body of health facilities to respective clinics. Verbal consent was obtained from study participants, and the purpose of this study was stated to all participants. Everybody participated voluntarily in this study.

## Author Contributions

**Conceptualization:** Tadesse Tolossa, Diriba Mulisa, Getahun Fetensa, Belayneh Mengist, Gebeyehu Abera, Tesfaye Shibiru.

**Data curation:** Diriba Ayala, Dejene Seyoum.

**Formal analysis:** Tadesse Tolossa, Bizuneh Wakuma, Ebisa Turi, Diriba Mulisa, Getahun Fetensa.

**Funding acquisition:** Bizuneh Wakuma, Diriba Mulisa, Diriba Ayala, Gebeyehu Abera, Emiru Merdassa Atomssa, Dejene Seyoum.

**Investigation:** Ebisa Turi, Getahun Fetensa, Belayneh Mengist, Gebeyehu Abera, Tesfaye Shibiru.

**Methodology:** Tadesse Tolossa, Bizuneh Wakuma, Ebisa Turi, Diriba Ayala, Dejene Seyoum.

**Project administration:** Getahun Fetensa, Belayneh Mengist, Gebeyehu Abera, Ayantu Getahun.

**Resources:** Tadesse Tolossa, Ebisa Turi, Diriba Mulisa, Emiru Merdassa Atomssa.

**Software:** Tadesse Tolossa, Bizuneh Wakuma, Belayneh Mengist, Tesfaye Shibiru.

**Supervision:** Bizuneh Wakuma, Diriba Ayala, Getahun Fetensa, Tesfaye Shibiru, Ayantu Getahun.

**Validation:** Diriba Mulisa, Getahun Fetensa, Belayneh Mengist, Gebeyehu Abera, Emiru Merdassa Atomssa.

**Visualization:** Tadesse Tolossa, Diriba Mulisa, Diriba Ayala, Gebeyehu Abera, Emiru Merdassa Atomssa, Dejene Seyoum.

**Writing – original draft:** Tadesse Tolossa, Bizuneh Wakuma, Ebisa Turi, Dejene Seyoum.

**Writing – review & editing:** Tadesse Tolossa, Bizuneh Wakuma, Ebisa Turi, Diriba Mulisa, Diriba Ayala, Getahun Fetensa, Emiru Merdassa Atomssa, Dejene Seyoum, Tesfaye Shibiru, Ayantu Getahun.

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
