## [Decision Letter · Decision Letter 0]

11 Jan 2022

PONE-D-21-21414Attitude and perception of health professionals towards COVID-19 vaccination and associated factors among health professionals found in Health facilities of Nekemte town, Western EthiopiaPLOS ONE

Dear Dr. Tolossa,

Thank you for submitting your manuscript to PLOS ONE. After careful consideration, we feel that it has merit but does not fully meet PLOS ONE’s publication criteria as it currently stands. Therefore, we invite you to submit a revised version of the manuscript that addresses the points raised during the review process.

The manuscript has been evaluated by two reviewers, and their comments are available below. Further comments to address are also included in the attached files and these need to be fully addressed.

The reviewers have raised a number of major concerns. They request improvements to the reporting of methodological aspects of the study, for example, providing more information on the tool used for data collection, the sample size calculation, and ensuring that the manuscript follows STROBE guidelines. In addition, the reviewers also note concerns about the quality of the written English and request that assistance is sought by a native English speaker with experience in scientific writing. 

Could you please carefully revise the manuscript to address all comments raised?

We look forward to receiving your revised manuscript.

Kind regards,

James Mockridge

Academic Editor

PLOS ONE

Journal Requirements:

- please provide the reference number for the ethical approval clearance

- please provide a completed STROBE checklist as an 'Other' file when resubmitting (see https://www.strobe-statement.org/checklists/)

- please clarify why verbal consent was provided and not written informed consent.

Reviewers' comments:

Reviewer's Responses to Questions

**Comments to the Author**

1. Is the manuscript technically sound, and do the data support the conclusions?

Reviewer #1: Yes

Reviewer #2: Partly

2. Has the statistical analysis been performed appropriately and rigorously? 

Reviewer #1: Yes

Reviewer #2: Yes

3. Have the authors made all data underlying the findings in their manuscript fully available?

Reviewer #1: Yes

Reviewer #2: Yes

4. Is the manuscript presented in an intelligible fashion and written in standard English?

Reviewer #1: Yes

Reviewer #2: No

5. Review Comments to the Author

Reviewer #1: Dear Author,

I have some comments and corrections. I added an MS Word file.

You should edit your manuscript with STROBE guideline.

You should give exact values in the result section.

You should checked your manuscript for typos.

Best wishes

Reviewer #2: Generally, there are lots of grammar issues, incohherence of writting methods and results,as a result unrserved effort is recommended to the authors to in crease the quality of the man uscript. All the comments are easily readable with Foxit reader in the manuscript.

6. PLOS authors have the option to publish the peer review history of their article (what does this mean?). If published, this will include your full peer review and any attached files.

Reviewer #1: No

Reviewer #2: No

---

## [Author Response · Author response to Decision Letter 0]

12 Jan 2022

Dear Academic Editor of PLOS ONE journal 

Dear Editor, this is regarding the manuscript PONE-D-21-21414 entitled as “Attitude and perception of health professionals towards COVID-19 vaccination and associated factors among health professionals found in Health facilities of Nekemte town, Western Ethiopia” submitted to PLOS ONE. Thanks for your time and consideration in editing and reviewing the manuscript. We have carefully read your comments and corrected inline of reviewer’s comments and suggestions. All comments raised were edited and incorporated in the main manuscript. Some of the changes were highlighted with yellow color in the manuscript. Here are the responses and elaborations for the comments!

Editor and Reviewer comments

The reviewers have raised a number of major concerns. They request improvements to the reporting of methodological aspects of the study, for example, providing more information on the tool used for data collection, the sample size calculation, and ensuring that the manuscript follows STROBE guidelines. In addition, the reviewers also note concerns about the quality of the written English and request that assistance is sought by a native English speaker with experience in scientific writing. 

Response: Dear editor, thank you very much, we have addressed these all issues (tool, sample size, language) in the revised manuscript.

Journal Requirements:

Response: Thank you Dear editor, we accepted your comment. All the revision was made in line with the journal requirements including the figure 

Response: Thank you dear, we have addressed this issue in the method part of the manuscript

Response: Thank you, we have included minimal dataset in the revised manuscript as supplementary data

- please provide the reference number for the ethical approval clearance

Response: Thank you dear, we have included reference number in the revised manuscript

- please provide a completed STROBE checklist as an 'Other' file when resubmitting (see https://www.strobe-statement.org/checklists/)

Response: We have included as supplementary file

- please clarify why verbal consent was provided and not written informed consent.

Response: Dear editor, thank you for your question; however, written consent was not obtained from respondents. In Ethiopia, written consent is only possible if a sample is needed from the patients and any invasive procedure is performed. Since, we were not received any blood sample and invasive procedure were not performed; only verbal consent was obtained from participants.

Reviewer #1: 

Dear Author,

I have some comments and corrections. I added an MS Word file.

You should edit your manuscript with STROBE guideline.

You should give exact values in the result section.

You should checked your manuscript for typos.

Response: Dear reviewer, Thank you for taking a time to review and edit our work thoroughly. Your effort in editing, and reviewing the overall document for its grammatical and vocabulary problem is very appreciable and valuable. Really, thank you in advance. We have tried to incorporate all your comments and corrections in the main manuscript. 

Reviewer #2: 

Generally, there are lots of grammar issues, incoherence of writing methods and results, as a result unreserved effort is recommended to the authors to increase the quality of the manuscript. All the comments are easily readable with Foxit reader in the manuscript.

Response: Dear Editor, we are very thankful for your important comment and we have tried to edit the grammatical flaws throughout the manuscript in its revised version. We have edited the spelling, grammatical errors, incomplete and poorly structured sentences throughout the manuscript. Now we believe the revised version is clean and clear enough to the readers.

1. There are lots of grammar issues in the manuscript, please correct it thoroughly?

Response: Thank you dear, we edited the grammatical error throughout the manuscript in its revised version

2. Two different results are reported in the abstract, which was associated with unfavorable attitude, was that good/poor knowledge ?

Response: Thank you dear, we have mentioned the the effect of knowledge on attitude towards the attitude in both result part and conclusion part, with the same message. In result part, it says good knowledge negatively associated with unfavorable attitude (implies having good knowledge about the vaccine could decrease the chance of poor attitude or good knowledge promote favorable attitude). In conclusion section it says poor knowledge positively affects unfavorable attitude, meaning lack of knowledge about the vaccine leads to poor attitude towards the vaccine. To make it clear, we have reported with the same statement in the revised manuscript.

3. One of your outcome, perception was not reported in the abstract section,why?

Response: Thank you dear, we have only one outcome variable (attitude). Perception was not the outcome variable, but we have used as an independent factor. To make it clear, we have removed perception from title in the revised manuscript.

4. Abbreviations/acronyms must be written in full form in first time writing? 

Response: kindly accepted your comment

5. There is also a study in Ethiopian population which stating the willingness of Ethiopian population to take the vaccine , it is better to cite it here?

Response: Thank you dear, we have cited studies done in Ethiopia in the revised manuscript.

6. Rlease, correct the punctuation ?

Response: Thank you dear, we have corrected it

7. Repetition of the word ''working'', please delete one of them?

Response: Corrected

8. Your total population is 800. so, why don’t you take either 50% of the population or correction formula, even you can take all of the professionals?

Response: Dear reviewer, we are grateful for this important question. Yes the total population is 800, and the estimated sample size was 439. We used p=50% to estimate the sample size. Our fear to take 50% of the total population was there is scientific background which says use 50% if the total population is 800. From the thumb rule of estimating sample size, it is possible take 50% if the total population is 500. So rather taking 50%, it better to calculate by taking p=50%. In addition, we did not used correction formula, because correction is simply used to save resources. So if we used correction formula the sample size would be less than 250 which could decrease the power of the study.

So, to have a scientific justification and increase the power of the study we used single population proportion formula.

9. Even your expected to give/revisit those health workers who were not on duty during your data collection, why?

Response: Dear, thank you for the question. Duty the urgency of the finding for the research communities we were not revisited the HCW who were on duty leave during data collection. However, we acknowledged this issue in the limitation section. 

10. Correct like '' the attitude of health professionals..''

Response: Thank you dear, we have corrected it.

11. Better to say attitude was takes as favorable when the over all score was greater than or equal to the≥ mean.

Response: corrected

12. Please add ''as'' after such.

Response: corrected

13. You should elaborate how do you develop your questions, adapted/adopted/validated/reliability?

Response: Thank you dear, we accepted your comment and incorporated in the revise manuscript

14. You should elaborate how do you develop your questions, adapted/adopted/validated/reliability?

Response: Thank you dear, we have accepted your comment and incorporated in the revised manuscript.

15. Was acceptability/willingness your objective or research question? factors must be fitted towards attitude/perception of COVID 19 vaccine?

Response: Dear reviewer, apologies, it is an technical error, and we have corrected it

16. How do you classify age like this? and use better symbols to make it more standard?

Response: Dear reviewer, thank you for the question. Of course, we need to categorize continuous data into categorical data based on the standards and references, but there was no standard to categorize age for health professionals. We simply categorized age into two categories by reviewing previous article.

17. It is not nearly less than quarter so your expression is not related with your report please correct it like ''only 19 or any other?

Response: Thank you dear, we have accepted your comment.

18. after saying half of, quarter of etc, it is better to add their respective number and percentage consequently?

Response: Thank you dear we have accepted your comment

19. Make it full table title ''where''? the table has grammar issues, correct them?

Response: Thank you dear, we have admitted your comment

20. grammar?

Response: corrected

21. Still you didn,t report the relationship b/n independent variables and perception towards COVID 19? or modify your title accordingly?

Response: Dear reviewer, thank you for this important comment, and we have removed perception from the title.

22. Since your interest variable unfavorable attitude it is better to start by unfavorable/poor attitude in your table?

Response: Thank you dear, we have accepted your comment and incorporated in the revised manuscript

23. You should report firstly to your research question/objective of your study, i.e ''attitude and perceptions of professionals towards COVID 19''? This report is the willingness of the participants?

Response: thank you dear, we have accepted your comment. But, 31.5% is not magnitude of vaccine hesitancy, but the attitude of health professional towards the vaccine hesisatncy. It is one item which was used to assess the attitude of health professionals. 

24. your reference system must be either Vancouver or Harvard?

Response: Kindly accepted your comment (Vancouver throughout the manuscript)

25. there is a study done in Ethiopia that showed the willingness to take the vaccine, so it is better to compare with it?

Response: Thank you dear we have reviewed and discussed our finding with previous studies conducted in Ethiopia

26. age < 30 years was significantly associated in your model, but you didn,t discuss it? in addition perceptions of the participants was not reported at all,so either modify your title or please adhere to your objective? 

Response: Thank you dear, we have incorporated your concerns in the revised manuscript

---

## [Decision Letter · Decision Letter 1]

24 Jan 2022

PONE-D-21-21414R1Attitude of health professionals towards COVID-19 vaccination and associated factors among health professionals, Western EthiopiaPLOS ONE

Dear Dr. Tolossa,

Thank you for submitting your manuscript to PLOS ONE. After careful consideration, we feel that it has merit but does not fully meet PLOS ONE’s publication criteria as it currently stands. Therefore, we invite you to submit a revised version of the manuscript that addresses the points raised during the review process.

TITLE:

Please add the study design in the title, as per the STROBE Guidelines. You might like to revise the title as follows: ‘Attitude of health professionals towards COVID-19 vaccination and associated factors among health professionals in Western Ethiopia: a cross sectional survey’

ABSTRACT:

Introduction:

Please delete ‘the’ before towards in this sentence: “…..health professionals the towards the COVID 19 vaccine in resource-limited settings like Ethiopia.”Please change ‘Hence, this study aimed’ to ‘The aim of study was’ in the following sentence:  “Hence, this study aimed to assess health professionals' attitudes and perceptions towards COVID 19 vaccine in West Ethiopia.”The authors have written ‘Western Ethiopia’ in the title and ‘West Ethiopia’ in the abstract. Please keep the same terminology in the whole manuscript so revise the text thoroughly.

Results:

Please delete ‘were’ in this sentence: “A total of 431 health professionals were participated…..”.Please change from: ‘The study indicates that…” to ‘The results indicated that..’Please report p values in the abstract.

Conclusion:

The following statement is not included and supported by the results hence it could be removed:  “Therefore, there is still a need to improve health professionals' knowledge of the COVID-19 vaccine by providing reliable information regarding vaccine safety, efficacy, and effectiveness.”

INTRODUCTION:

Please delete ‘and’ before ‘the’ in this sentence “…., and the COVID-19…’Please delete either ‘coronavirus’ or  ‘COVID-19’from this sentence: “According to the worldometer report, the coronavirus COVID-19…”.Please revise this sentence: “More than 170 million cases, and 3.5 million deaths happened due to COVID-19 [1, 2].” as “More than 170 million cases, and 3.5 million deaths have happened due to COVID-19, as of (add date/month/year) [1, 2].Please delete ‘health care system challenges;, which is given twice in this sentence: “The pandemic brought the double burden in developing countries already overwhelmed by the health care system challenges already overwhelmed by the health care system challenges [3].”The following information has become old so it could be omitted and you can name a few COVID vaccines that are being used globally or locally. “Many vaccines started to arise around 2020; there are hundreds of candidate vaccines [7]. As of June 10, 2021, from 287 candidate vaccines, 102 are in the clinical phase, 185 are in the preclinical phase [8]. From these, WHO has listed the Pfizer/BioNTech, Astrazeneca-SK Bio,Serum Institute of India, Janssen and Moderna vaccines for emergency use [6, 7].Please delete ‘hope of’ from this sentence: Even though people of the world were eagerly waiting for the hope of vaccine development…”Please delete ‘A’ before ‘studies’ in the following sentence: “A studies conducted in different regions of Ethiopia, ….”Please revise “Several evidence indicates’ to ‘Several studies indicate’….Please delete ‘now’ in this sentence:  “…..9651 deaths till now, May 30, 2021 [1].”

METHODS:

Please change ‘found’ to ‘located’ in this sentence: “This study was conducted in health institutions found in Nekemte town.”Please add a full stop / period (.) at the end of this sentence: More than 800 health professionals are found in Nekemte townSample size: please double check the sample size with15% non-response, as my calculation is a little higher than 439 reported.Please report who was included/considered as ‘health professionals’. Was it every body working in hospitals or only doctors, nurses and AHPs.Variable mean: The authors report that items were measured on 5 point Likert scale  and “Attitude was takes as “favorable” when the overall score was greater than or equal to the mean and less than mean value was rated as an "unfavorable attitude" towards COVID-19 vaccine.” Could you please what were scores for each point of the scale and what was the mean value of the scale scores.Could you please either refer to relevant table or report 10 items used to measure the attitude and report whether these items were adapted from an earlier survey or developed at home. Please report sources if adapted.Please either refer to relevant table or report 5 items used to assess the knowledge of health professionals towards the COVID-19 vaccine. Also, report whether these items were adapted from an earlier survey or developed at home. Please report sources if adapted. Please either refer to relevant table or report 5 items used to measure the perception and report whether these items were adapted from an earlier survey or developed at home. Please report sources if adapted. Could you also report how yes or no were scored.There is repetition of reporting language in which questionnaire was developed. “The tool was designed and distributed to respondents in English language…”.  AND “The questionnaire was prepared in English,..”. Please avoid repetition and revise the text.Please report the acceptable level of Cronbach’s alpha in the following sentence: “Cronbach’s alpha was used to assess the reliability of the tool.”Please refer questionnaire included as supplementary material.

Software:

Software: please report citations and references for Epidata and STATA software used.

Ethics:

Please report the data and number of ethics approval by the Ethics Review Board.

RESULTS

In the methods section it is reported that the questionnaire was given to health professionals for self- completion while the results section reports that the health professionals were interviewed. Please provide the correct information about how and who completed the questionnaires Either respondents or researchers or both.You might like to delete ‘Again in this study’ in this sentence: “Again in this study, greater than half of the participants were vaccinated against coronavirus 240 (55.68%).” You might like to remove ‘In this study’ from this sentence: “In this study, almost half 210 (48.72 %) of the participants have poor attitude toward COVID-19 vaccination.”Please change ‘discovered’ to ‘developed’ in the following sentence: “Regarding the effectiveness of the newly discovered COVID-19 vaccination,..”Table 3: please check the number of respondents for item 3 “Do you think that covid vaccine is mandatory for health care workers?” because the total number is higher than 341 (Yes 262, 60.79%, No 262, 60.79% and I don’t know 145, 33.64%.Please double check data (counts and %) in all tables.

DISCUSSION

Please revise/change ‘Works of literature are reporting as vaccine..” to ‘Literature shows that as vaccine…Please change ‘the worries of the peoples’ to ‘the worries of the people…”Please avoid reporting results in the discussion section like ‘AOR=0.38(95%CI: 0.22, 0.64)).’

REFERNCES:

Please report abbreviations of journal names as reported in the PubMed, if available. Please submit your revised manuscript by Mar 10 2022 11:59PM. If you will need more time than this to complete your revisions, please reply to this message or contact the journal office at plosone@plos.org. Please include the following items when submitting your revised manuscript:A rebuttal letter that responds to each point raised by the academic editor and reviewer(s). You should upload this letter as a separate file labeled 'Response to Reviewers'.A marked-up copy of your manuscript that highlights changes made to the original version. You should upload this as a separate file labeled 'Revised Manuscript with Track Changes'.An unmarked version of your revised paper without tracked changes. You should upload this as a separate file labeled 'Manuscript'.If applicable, we recommend that you deposit your laboratory protocols in protocols.io to enhance the reproducibility of your results. Protocols.io assigns your protocol its own identifier (DOI) so that it can be cited independently in the future. For instructions see: https://journals.plos.org/plosone/s/submission-guidelines#loc-laboratory-protocols. Additionally, PLOS ONE offers an option for publishing peer-reviewed Lab Protocol articles, which describe protocols hosted on protocols.io. Read more information on sharing protocols at https://plos.org/protocols?utm_medium=editorial-email&utm_source=authorletters&utm_campaign=protocols.

We look forward to receiving your revised manuscript.

Kind regards,

Syed Ghulam Sarwar Shah, M.B.B.S., M.A., M.Sc., Ph.D.

Academic Editor

PLOS ONE

Journal Requirements:

Additional Editor Comments:

Please address the following issues.

TITLE:

Please add the study design in the title, as per the STROBE Guidelines. You might like to revise the title as follows: ‘Attitude of health professionals towards COVID-19 vaccination and associated factors among health professionals in Western Ethiopia: a cross sectional survey’

ABSTRACT:

Introduction:

1. Please delete ‘the’ before towards in this sentence: “…..health professionals the towards the COVID 19 vaccine in resource-limited settings like Ethiopia.”

2. Please change ‘Hence, this study aimed’ to ‘The aim of study was’ in the following sentence: “Hence, this study aimed to assess health professionals' attitudes and perceptions towards COVID 19 vaccine in West Ethiopia.”

3. The authors have written ‘Western Ethiopia’ in the title and ‘West Ethiopia’ in the abstract. Please keep the same terminology in the whole manuscript so revise the text thoroughly.

Results:

4. Please delete ‘were’ in this sentence: “A total of 431 health professionals were participated…..”.

5. Please change from: ‘The study indicates that…” to ‘The results indicated that..’

6. Please report p values in the abstract.

Conclusion:

7. The following statement is not included and supported by the results hence it could be removed: “Therefore, there is still a need to improve health professionals' knowledge of the COVID-19 vaccine by providing reliable information regarding vaccine safety, efficacy, and effectiveness.”

INTRODUCTION:

8. Please delete ‘and’ before ‘the’ in this sentence “…., and the COVID-19…’

9. Please delete either ‘coronavirus’ or ‘COVID-19’from this sentence: “According to the worldometer report, the coronavirus COVID-19…”.

10. Please revise this sentence: “More than 170 million cases, and 3.5 million deaths happened due to COVID-19 [1, 2].” as “More than 170 million cases, and 3.5 million deaths have happened due to COVID-19, as of (add date/month/year) [1, 2].

11. Please delete ‘health care system challenges;, which is given twice in this sentence: “The pandemic brought the double burden in developing countries already overwhelmed by the health care system challenges already overwhelmed by the health care system challenges [3].”

12. The following information has become old so it could be omitted and you can name a few COVID vaccines that are being used globally or locally. “Many vaccines started to arise around 2020; there are hundreds of candidate vaccines [7]. As of June 10, 2021, from 287 candidate vaccines, 102 are in the clinical phase, 185 are in the preclinical phase [8]. From these, WHO has listed the Pfizer/BioNTech, Astrazeneca-SK Bio,Serum Institute of India, Janssen and Moderna vaccines for emergency use [6, 7].

13. Please delete ‘hope of’ from this sentence: Even though people of the world were eagerly waiting for the hope of vaccine development…”

14. Please delete ‘A’ before ‘studies’ in the following sentence: “A studies conducted in different regions of Ethiopia, ….”

15. Please revise “Several evidence indicates’ to ‘Several studies indicate’….

16. Please delete ‘now’ in this sentence: “…..9651 deaths till now, May 30, 2021 [1].”

METHODS:

17. Please change ‘found’ to ‘located’ in this sentence: “This study was conducted in health institutions found in Nekemte town.”

18. Please add a full stop / period (.) at the end of this sentence: More than 800 health professionals are found in Nekemte town

19. Sample size: please double check the sample size with15% non-response, as my calculation is a little higher than 439 reported.

20. Please report who was included/considered as ‘health professionals’. Was it every body working in hospitals or only doctors, nurses and AHPs.

21. Variable mean: The authors report that items were measured on 5 point Likert scale and “Attitude was takes as “favorable” when the overall score was greater than or equal to the mean and less than mean value was rated as an "unfavorable attitude" towards COVID-19 vaccine.” Could you please what were scores for each point of the scale and what was the mean value of the scale scores.

22. Could you please either refer to relevant table or report 10 items used to measure the attitude and report whether these items were adapted from an earlier survey or developed at home. Please report sources if adapted.

23. Please either refer to relevant table or report 5 items used to assess the knowledge of health professionals towards the COVID-19 vaccine. Also, report whether these items were adapted from an earlier survey or developed at home. Please report sources if adapted.

24. Please either refer to relevant table or report 5 items used to measure the perception and report whether these items were adapted from an earlier survey or developed at home. Please report sources if adapted. Could you also report how yes or no were scored.

25. There is repetition of reporting language in which questionnaire was developed. “The tool was designed and distributed to respondents in English language…”. AND “The questionnaire was prepared in English,..”. Please avoid repetition and revise the text.

26. Please report the acceptable level of Cronbach’s alpha in the following sentence: “Cronbach’s alpha was used to assess the reliability of the tool.”

27. Please refer questionnaire included as supplementary material.

Software:

28. Software: please report citations and references for Epidata and STATA software used.

Ethics:

29. Please report the data and number of ethics approval by the Ethics Review Board.

RESULTS

30. In the methods section it is reported that the questionnaire was given to health professionals for self- completion while the results section reports that the health professionals were interviewed. Please provide the correct information about how and who completed the questionnaires Either respondents or researchers or both.

31. You might like to delete ‘Again in this study’ in this sentence: “Again in this study, greater than half of the participants were vaccinated against coronavirus 240 (55.68%).”

32. You might like to remove ‘In this study’ from this sentence: “In this study, almost half 210 (48.72 %) of the participants have poor attitude toward COVID-19 vaccination.”

33. Please change ‘discovered’ to ‘developed’ in the following sentence: “Regarding the effectiveness of the newly discovered COVID-19 vaccination,..”

34. Table 3: please check the number of respondents for item 3 “Do you think that covid vaccine is mandatory for health care workers?” because the total number is higher than 341 (Yes 262, 60.79%, No 262, 60.79% and I don’t know 145, 33.64%.

35. Please double check data (counts and %) in all tables.

DISCUSSION

36. Please revise/change ‘Works of literature are reporting as vaccine..” to ‘Literature shows that as vaccine…

37. Please change ‘the worries of the peoples’ to ‘the worries of the people…”

38. Please avoid reporting results in the discussion section like ‘AOR=0.38(95%CI: 0.22, 0.64)).’

REFERENCES:

39. Please report abbreviations of journal names as reported in the PubMed, if available.

Reviewers' comments:

Reviewer's Responses to Questions

**Comments to the Author**

1. If the authors have adequately addressed your comments raised in a previous round of review and you feel that this manuscript is now acceptable for publication, you may indicate that here to bypass the “Comments to the Author” section, enter your conflict of interest statement in the “Confidential to Editor” section, and submit your "Accept" recommendation.

Reviewer #1: All comments have been addressed

Reviewer #2: All comments have been addressed

2. Is the manuscript technically sound, and do the data support the conclusions?

Reviewer #1: Yes

Reviewer #2: Yes

3. Has the statistical analysis been performed appropriately and rigorously? 

Reviewer #1: Yes

Reviewer #2: Yes

4. Have the authors made all data underlying the findings in their manuscript fully available?

Reviewer #1: Yes

Reviewer #2: Yes

5. Is the manuscript presented in an intelligible fashion and written in standard English?

Reviewer #1: Yes

Reviewer #2: Yes

6. Review Comments to the Author

Reviewer #1: Dear Author,

Thank you for all corrections.

I do not have an additional recommendation.

It is publishable work for me.

Best wishes

Reviewer #2: the title of figure 2;"is the corna virus vaccine has side effects?" must be wirriten in better way

7. PLOS authors have the option to publish the peer review history of their article (what does this mean?). If published, this will include your full peer review and any attached files.

Reviewer #1: No

Reviewer #2: No

---

## [Author Response · Author response to Decision Letter 1]

18 Feb 2022

Dear Academic Editor of PLOS ONE journal 

Dear Editor, this is regarding the manuscript PONE-D-21-21414 entitled as “Attitude of health professionals towards COVID-19 vaccination and associated factors among health professionals found in Health facilities of Nekemte town, Western Ethiopia” submitted to PLOS ONE. Thanks for your time and consideration in editing and reviewing the manuscript. We have carefully read your comments and corrected inline of reviewer’s comments and suggestions. All comments raised were edited and incorporated in the main manuscript. Some of the changes were highlighted with yellow color in the manuscript. Here are the responses and elaborations for the comments!

TITLE

Please add the study design in the title, as per the STROBE Guidelines. You might like to revise the title as follows: ‘Attitude of health professionals towards COVID-19 vaccination and associated factors among health professionals in Western Ethiopia: a cross sectional survey’.

Response: revised

ABSTRACT

Introduction:

1. Please delete ‘the’ before towards in this sentence: “…..health professionals the towards the COVID 19 vaccine in resource-limited settings like Ethiopia.”

Response: corrected 

2. Please change ‘Hence, this study aimed’ to ‘The aim of study was’ in the following sentence: “Hence, this study aimed to assess health professionals' attitudes and perceptions towards COVID 19 vaccine in West Ethiopia.”

Response: corrected

3. The authors have written ‘Western Ethiopia’ in the title and ‘West Ethiopia’ in the abstract. Please keep the same terminology in the whole manuscript so revise the text thoroughly.

Response: corrected 

Results:

4. Please delete ‘were’ in this sentence: “A total of 431 health professionals were participated…..”.

Response: corrected

5. Please change from: ‘The study indicates that…” to ‘The results indicated that..’

Response: corrected

6. Please report p values in the abstract.

Response: corrected

Conclusion:

7. The following statement is not included and supported by the results hence it could be removed: “Therefore, there is still a need to improve health professionals' knowledge of the COVID-19 vaccine by providing reliable information regarding vaccine safety, efficacy, and effectiveness.”

Response: corrected

INTRODUCTION:

8. Please delete ‘and’ before ‘the’ in this sentence “…., and the COVID-19…’

Response: corrected

9. Please delete either ‘coronavirus’ or ‘COVID-19’from this sentence: “According to the worldometer report, the coronavirus COVID-19…”.

Response: corrected

10. Please revise this sentence: “More than 170 million cases, and 3.5 million deaths happened due to COVID-19 [1, 2].” as “More than 170 million cases, and 3.5 million deaths have happened due to COVID-19, as of (add date/month/year) [1, 2].

Response: corrected

11. Please delete ‘health care system challenges;, which is given twice in this sentence: “The pandemic brought the double burden in developing countries already overwhelmed by the health care system challenges already overwhelmed by the health care system challenges [3].”

 Response: corrected

12. The following information has become old so it could be omitted and you can name a few COVID vaccines that are being used globally or locally. “Many vaccines started to arise around 2020; there are hundreds of candidate vaccines [7]. As of June 10, 2021, from 287 candidate vaccines, 102 are in the clinical phase, 185 are in the preclinical phase [8]. From these, WHO has listed the Pfizer/BioNTech, Astrazeneca-SK Bio,Serum Institute of India, Janssen and Moderna vaccines for emergency use [6, 7].

Response: corrected

13. Please delete ‘hope of’ from this sentence: Even though people of the world were eagerly waiting for the hope of vaccine development…”

Response: corrected

14. Please delete ‘A’ before ‘studies’ in the following sentence: “A studies conducted in different regions of Ethiopia,….”

Response: corrected

15. Please revise “Several evidence indicates’ to ‘Several studies indicate’….

Response: corrected

16. Please delete ‘now’ in this sentence: “…..9651 deaths till now, May 30, 2021 [1].”

Response: corrected

METHODS:

17. Please change ‘found’ to ‘located’ in this sentence: “This study was conducted in health institutions found in Nekemte town.”

Response: corrected

18. Please add a full stop / period (.) at the end of this sentence: More than 800 health professionals are found in Nekemte town

Response: corrected

19. Sample size: please double check the sample size with15% non-response, as my calculation is a little higher than 439 reported.

Response: Thank you dear, it is a editorial error and we have corrected it

20. Please report who was included/considered as ‘health professionals’. Was it every body working in hospitals or only doctors, nurses and AHPs.

Response: Thank you dear, health professionals were all health care workers in Nekemte town regardless of their profession.

21. Variable mean: The authors report that items were measured on 5 point Likert scale and “Attitude was takes as “favorable” when the overall score was greater than or equal to the mean and less than mean value was rated as an "unfavorable attitude" towards COVID-19 vaccine.” Could you please what were scores for each point of the scale and what was the mean value of the scale scores.

Response: revised in the updated version of manuscript

22. Could you please either refer to relevant table or report 10 items used to measure the attitude and report whether these items were adapted from an earlier survey or developed at home. Please report sources if adapted.

Response: revised in the updated version of manuscript

23. Please either refer to relevant table or report 5 items used to assess the knowledge of health professionals towards the COVID-19 vaccine. Also, report whether these items were adapted from an earlier survey or developed at home. Please report sources if adapted.

Response: revised in the updated version of manuscript

24. Please either refer to relevant table or report 5 items used to measure the perception and report whether these items were adapted from an earlier survey or developed at home. Please report sources if adapted. Could you also report how yes or no were scored.

Response: revised in the updated version of manuscript

25. There is repetition of reporting language in which questionnaire was developed. “The tool was designed and distributed to respondents in English language…”. AND “The questionnaire was prepared in English,..”. Please avoid repetition and revise the text.

Response: revised in the updated version of manuscript

26. Please report the acceptable level of Cronbach’s alpha in the following sentence: “Cronbach’s alpha was used to assess the reliability of the tool.”

Response: included in the revised version of manuscript

27. Please refer questionnaire included as supplementary material.

Response: Hyperlinked

Software:

28. Software: please report citations and references for Epidata and STATA software used.

Response: Cited

Ethics:

29. Please report the data and number of ethics approval by the Ethics Review Board.

Response: ethical clearance number is included in the revised manuscript

RESULTS

30. In the methods section it is reported that the questionnaire was given to health professionals for self- completion while the results section reports that the health professionals were interviewed. Please provide the correct information about how and who completed the questionnaires Either respondents or researchers or both.

Response: Thank you dear, we have corrected it

31. You might like to delete ‘Again in this study’ in this sentence: “Again in this study, greater than half of the participants were vaccinated against coronavirus 240 (55.68%).”

Response: corrected 

32. You might like to remove ‘In this study’ from this sentence: “In this study, almost half 210 (48.72 %) of the participants have poor attitude toward COVID-19 vaccination.”

Response: corrected 

33. Please change ‘discovered’ to ‘developed’ in the following sentence: “Regarding the effectiveness of the newly discovered COVID-19 vaccination,..”

Response: corrected 

34. Table 3: please check the number of respondents for item 3 “Do you think that covid vaccine is mandatory for health care workers?” because the total number is higher than 341 (Yes 262, 60.79%, No 262, 60.79% and I don’t know 145, 33.64%.

Response: corrected 

35. Please double check data (counts and %) in all tables.

Response: corrected 

DISCUSSION

36. Please revise/change ‘Works of literature are reporting as vaccine..” to ‘Literature shows that as vaccine…

Response: corrected 

37. Please change ‘the worries of the peoples’ to ‘the worries of the people…”

Response: corrected 

38. Please avoid reporting results in the discussion section like ‘AOR=0.38(95%CI: 0.22, 0.64)).’

Response: corrected 

REFERNCES:

39. Please report abbreviations of journal names as reported in the PubMed, if available.

Response: corrected

---

## [Editor Report · Decision Letter 2]

23 Feb 2022

Attitude of health professionals towards COVID-19 vaccination and associated factors among health professionals, Western Ethiopia: a cross-sectional survey

PONE-D-21-21414R2

Dear Dr. Tolossa,

We’re pleased to inform you that your manuscript has been judged scientifically suitable for publication and will be formally accepted for publication once it meets all outstanding technical requirements.

Kind regards,

Syed Ghulam Sarwar Shah, M.B.B.S., M.A., M.Sc., Ph.D.

Academic Editor

PLOS ONE

Additional Editor Comments (optional):

The authors need to report journal names in their abbreviated form, which was suggested by the Academic editor in his last report. There are similar articles from the same region of the same country i.e. Ethiopia. Possible duplication or plagiarism may be checked before the acceptance decision is conveyed to the authors.
---

## [Editor Report · Acceptance letter]

28 Feb 2022

PONE-D-21-21414R2 

Attitude of health professionals towards COVID-19 vaccination and associated factors among health professionals, Western Ethiopia: a cross-sectional survey 

Dear Dr. Tolossa:

I'm pleased to inform you that your manuscript has been deemed suitable for publication in PLOS ONE. Congratulations! Your manuscript is now with our production department. 

Kind regards, 

on behalf of

Dr. Syed Ghulam Sarwar Shah 

Academic Editor

PLOS ONE